# Assessment of Agricultural Water Resources Sustainability in Arid Regions Using Virtual Water Concept: Case of South Khorasan Province, Iran

**Ehsan Qasemipour** [1] and **Ali Abbasi** [1,2,*]

1   Department of Civil Engineering, Faculty of Engineering, Ferdowsi University of Mashhad, Mashhad 9177948974, Iran; ehsanqasemi14@gmail.com
2   Faculty of Civil Engineering and Geosciences, Water Resources Section, Delft University of Technology, Stevinweg 1, 2628 CN Delft, The Netherlands
*   Correspondence: a.abbasi@tudelft.nl or aabbasi@um.ac.ir; Tel.: +31-15-2781029

**Abstract:** Cropping pattern plays an important role in providing food and agricultural water resources sustainability, especially in arid regions in which the concomitant socioeconomic dangers of water shortage would be inevitable. In this research, six indices are applied to classify 37 cultivated crops according to Central Product Classification (CPC). The respective 10-year data (2005–2014) were obtained from Agricultural Organization of South Khorasan (AOSKh) province. The water footprint concept along with some economic indicators are used to assess the water use efficiency. Results show that blue virtual water contributes to almost 99 percent of Total Virtual Water (TVW). In this occasion that an increasing pressure is exerted on groundwater resources, improper pattern of planting crops has to be beyond reproach. The improper cropping pattern in the study area led to the overuse of $346 \times 10^6$ m$^3$ of water annually. More specifically, cereals cultivation was neither environmentally nor economically sustainable and since they accounted for the largest share of water usage at the province level, importing them should be considered as an urgent priority. Vegetable cultivation could be further increased—instead of other water-intensive crops such as fruits—at the province level, not only due to their low TVW, but also to their higher financial output.

**Keywords:** water footprint; water use efficiency; virtual water trade; water self-sufficiency; sustainability; arid regions

## 1. Introduction

Population booming along with climate change and economic miracle has posed serious dangers to water supply deprivation. In this situation, the uneven temporal and spatial distribution of water resources in (semi-)arid regions exacerbate the environmental problems originating from overexploitation of groundwater aquifers. Declining groundwater levels, dried lakes, and land subsidence show clear signs of water scarcity (WS) [?   ?]. Inexorable linkage between the water and food has consequently attracted considerable attention from researchers of different disciplines in recent years [?   ?]. Agriculture is the major user of water and, therefore, has a significant effect on water resources around the world, especially in arid and semi-arid regions [?   ?]. Human actions exert a lasting influence not only on water-cycle, and in turn, in providing food security, but also on the natural ecosystems, social well-being, and the economy of an area. What is more, we should be able to feed a global population of more than 9 billion people whom are likely to live in the next 40 years [?].

The solution lies in changing the production practices and modifying the cultivation patterns. In order to do so, Hoekstra and Hung (2002) [?] developed a new concept named virtual water as

an indicator to measure both two rainfall and groundwater resources used in the production chain (namely green and blue water footprint, respectively). Using this concept is now globally recognized as an effective strategy in order to save water regarding sustainability of the water resources. For example, the UK, Brazil, China and the United States have applied this strategy to improve their water resources sustainability [**? ? ? ?** ].

Merely a few comprehensive assessments are carried out in Iran regarding the evaluation of water resource management and sustainability using the water footprint concept as well as other indices like Water Self-sufficiency [**? ?** ]. This is while other studies often focused their attention on calculating different components of water footprint in some particular agricultural products, rather than on the sustainability assessment of the water resource allocation to agricultural practices [**? ?** ].

The aim of this study is to evaluate the sustainability of water resources in the South Khorasan province. Prolonged drought granted permission to the farmers to exploit groundwater supplies during the drought. According to the Regional Water Company of South Khorasan (RWC-SKh), more than 90% of the available water in the province has been used in traditional agriculture, therefore, regarding this high consumption, the role of proper water allocation in cultivation patterns in this region should be addressed. Different crops are harvested in the province and recent droughts during 2005–now have not reduced the water withdrawals, although the harvested area decreased by 23% [**?** ]. This is mainly due to the improper reallocations of cultivation lands planned by the authorities. In this paper, the sustainability of the crop production in the period of 2005–2014 is investigated applying the water footprint concept and measuring the virtual water flows in the province. Some indicators, including Water Self-Sufficiency (WSS) and average revenue per hectare, are applied in this study to evaluate the province situation regarding the above-mentioned issues. WSS is an indicator which reflects the level a region is independent of other regions to meet its required water for producing its foods locally [**?** ]. This indicator is used not only for international evaluation of water resources [**? ?** ], but also for local water resource management [**?** ]. Different sets of meteorological and agricultural data have been analyzed and applied to quantify the water footprint of different crops using the CROPWAT model [**?** ].

## 2. Materials and Methods

### 2.1. Study Area

South Khorasan province is located in the east of Iran and lies between $30°15'$ N to $34°03'$ N latitudes and $57°43'$ E to $61°04'$ E longitudes with almost 151,193 square km area and 768,898 inhabitants. Birjand, Boshrooye, Darmian, Ferdows, Khusf, Nehbandan, Qaen, Sarayan, Sarbishe, Tabas, and Zirkooh are the 11 counties of the province as shown in Figure **??**. This province accounts for almost 9 percent of Iran's territory and is the third biggest province in the whole country. Temperature is ranging from $-13.57$ °C to 46.83 °C with a dry and hot summer. Spatial pattern of precipitation in this region varies between 89 mm in Tabas to 186 mm in Qaen. Temporal distribution of precipitation is even more uneven. Most of the annual rainfall occurs from January to April. As the province is located near one of the hottest deserts in the world (Dasht-e Lut), low relative humidity is a common feature of this region with 10 to 58 percent. The east (Tabas) and south (Nehbandan) part of the province are the driest parts and Qaen in the north has the highest humidity. The dominant wind direction is north to south with speed of up to 6.5 m/s. Due to the high temperature values during the year alongside the moderate wind speed values, the evapotranspiration amount is high ranging from 964 mm to 1552 mm per year in the study region.

10-year climate data (2005–2014) was obtained from South Khorasan Province Metrological Organization (SKhMet) [**?** ]. The collected data includes monthly average maximum and minimum temperature, wind speed, sunshine hours, precipitation, and relative humidity measured in one station in each county. Agricultural data consists of harvesting area, crop production, crop yield and planting and harvesting dates obtained from AOSKh [**?** ].

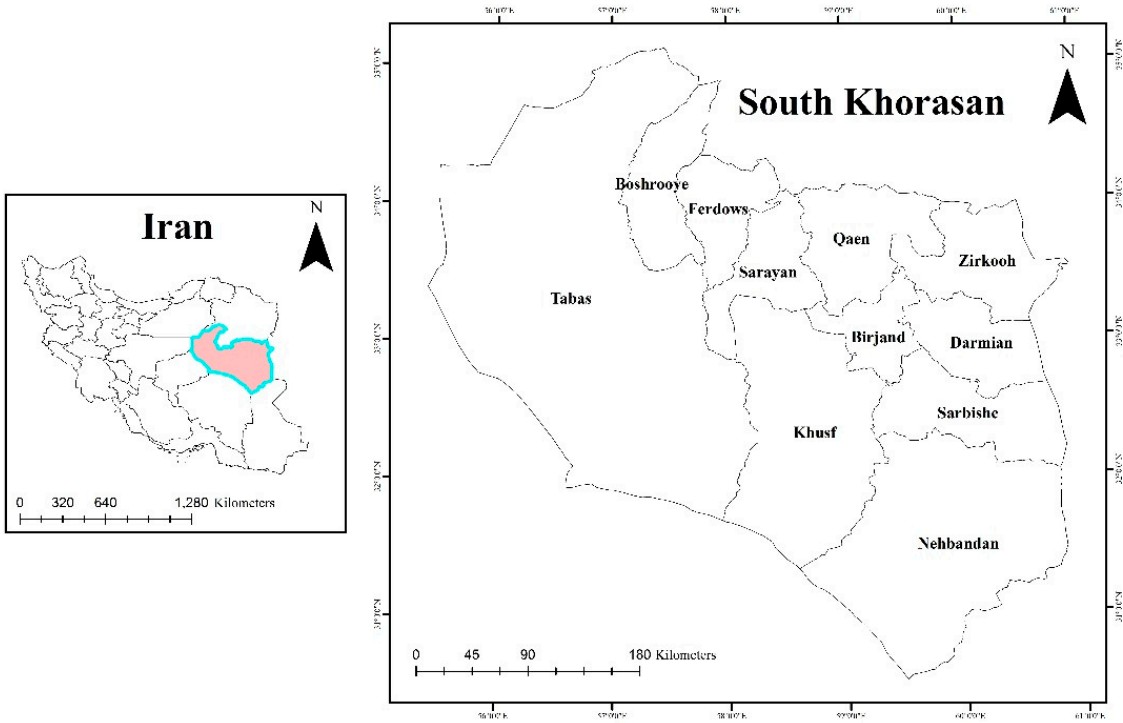

**Figure 1.** Map of the study area, South Khorasan and its 11 counties.

In this section, the virtual water content of main 37 cultivated crops in the South Khorasan province is investigated. The water footprint of crop production is the volume of water used in the production chain, which is a composition of precipitation (green WF), irrigation resources (blue WF), and the amount of water is needed to assimilate the pollution in order to maintain the existing ambient water quality within the standard levels (grey WF). In addition, considering the concept of virtual water, the virtual water trade in the province is analyzed. To analyze the water resource management of the province, an annual temporal scale, county spatial level, and two sources of water (i.e., rainfall and irrigation) is applied. It should be added that this study considers only the amount of water is used in land, without considering the grey WF, to produce unprocessed crops not the total amount of water used in their production and consumption chain. As a result, the virtual water content term is used to assess the agricultural sustainability of the water resources of the province, instead of water footprint term. By using the CROPWAT model, it is possible to distinguish between the rainfall (green) and the irrigation (blue) sources of water used to produce one mass unit of a crop, and therefore, calculating the net virtual water flows, water self-sufficiency and WS would be possible for each share. In this study, the VWC of a crop is estimated as the sum of the green and blue VWC only.

Considering exporting water in the virtual form and its concomitant dangers according to the limited water resources in some regions is undeniable. In arid regions, when the available water resources are limited such as South Khorasan province, this challenge is more critical and therefore, assessment of virtual water concept should be considered in more details. If this concept applied correctly, the water shortage could be alleviated in the study region.

In this study, initially, the procedures of figuring out the virtual water content of unprocessed agricultural crops are described, and then the water footprints of these crops are calculated. Regarding the calculated water footprints and following the procedures depicted in Figure **??**, it is analyzed how the water footprint of unprocessed crops cultivated in the region pose the main threat to sustainable development of the province. To reach this aim, a variety of practical indices including water self-sufficiency (WSS) and water scarcity (WS) indices are defined to show how overexploitation of water resources can endanger not only the environmental but also the socioeconomic sustainability.

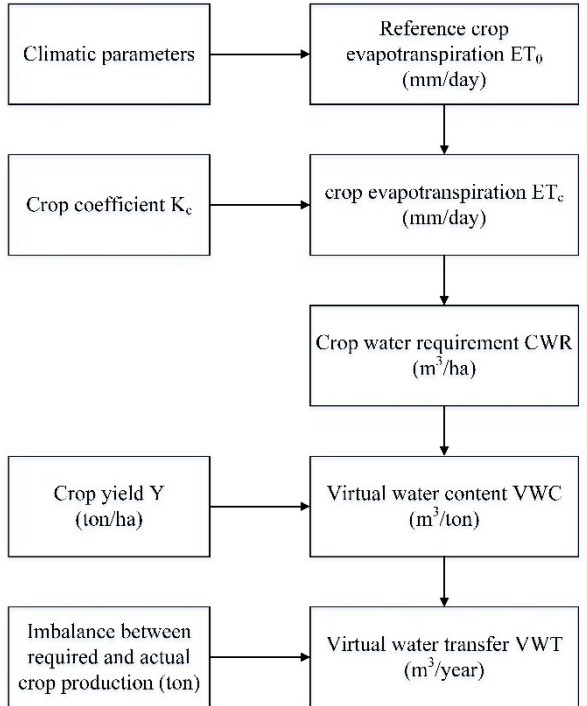

**Figure 2.** Procedures in calculating the virtual water flows (adopted from Liu et al. (2015) [? ]).

*2.2. Virtual Water Content Estimates*

A crop's Virtual Water Content (VWC) is the ratio of total crop evapotranspiration (ET, in $m^3$/ha) to the crop yield (Y, in ton/ha) as described by Hoekstra and Hung [? ]:

$$VWC = \frac{ET}{Y} \tag{1}$$

The selected 37 crops in this research account for almost 80 percent of cultivated crops in South Khorasan [? ]. For these cultivated crops in the region, the contribution of different types of water resource (i.e., green and blue) to the total crop evapotranspiration are distinguished in order to measure the participation of all sources in the province. Comparison of these contribution percentages will guide the local authorities through the dominant water resources in the region to make more practical decisions to control them.

*2.3. Crop Evapotranspiration*

Evapotranspiration from the two sources of available water in the region (i.e., rainfall and irrigation) is calculated as follows:

$$ET = K_c \times ET_0 \tag{2}$$

where ET refers to evapotranspiration (mm). To calculate ET from both the green and blue water resources the CROPWAT model is used in this study [? ]. The CROPWAT model is a decision support tool developed by the Land and Water Development Division of Food and Agriculture Organization (FAO) to help water authorities and decision makers determine crop water requirements [? ]. This model calculates the reference crop evapotranspiration using the soil moisture balance with the Penman-Monteith method. Metrological data such as maximum and minimum air temperature, precipitation, relative humidity, wind speed, and sunshine hours are considered as input variables

in this model. The reference crop evapotranspiration is solely affected by climatic parameters and is defined as the evapotranspiration from a reference surface and is quantified as follows [**?** ]:

$$ET_0 = \frac{0.408\Delta(R_n - G) + \gamma\frac{900(e_a - e_d)}{T+273}}{\Delta + \gamma(1 + 0.34\,U_2)} \tag{3}$$

in which $R_n$ is the net radiation at the grass surface ($MJ/m^2 \cdot h$), G is the soil heat flux density ($MJ/m^2 \cdot h$), $\Delta$ is the slope of the saturated vapor pressure curve at temperature T ($kPa/°C$), $\gamma$ is psychrometric constant ($kPa/°C$), T is mean hourly air temperature ($°C$), $U_2$ is averaged hourly wind speed at height of 2 m above the ground surface (m/s), and $e_a$ and $e_d$ are the saturation and actual vapor pressure, respectively (kPa).

The crop coefficients ($K_c$) are computed on a daily basis during the growing period of crops. Following the proposed approach by Allen et al. [**?** ], the total crop evapotranspiration is determined by summation the results of multiplying daily $K_c$ by reference evapotranspiration ($ET_0$) as shown in Equation (2). The evapotranspiration rate (in $m^3$ per hectare) for each county is depicted in Figure **??**. One of the capabilities of the CROPWAT model is distinguishing between evapotranspiration from green and blue water resources.

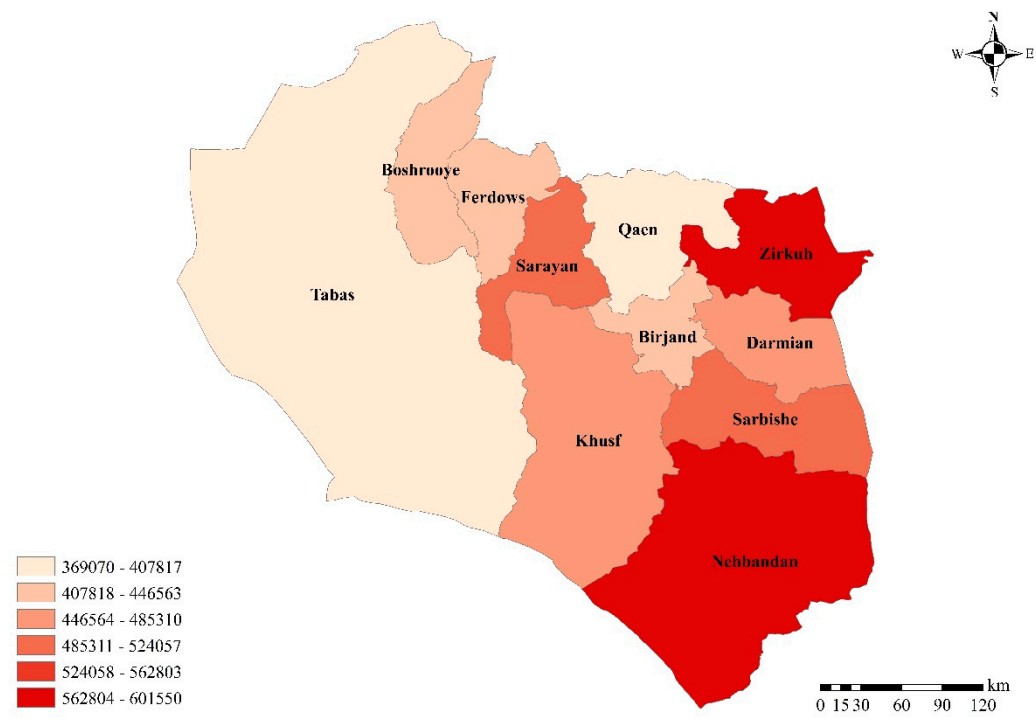

**Figure 3.** Evapotranspiration rate in South Khorasan province ($m^3$ per hectare).

It is worth mentioning that, crop coefficients for each crop have undergone some changes in their quantity in each county as some county's relative humidity and wind speed do not observe the minimum conditions (RH = 45%, $K_c$ = 2 m/s) using the methodology described in Allen et al. [**?** ].

The Virtual Water Content (VWC) of a crop determines how much water is embodied in an unprocessed crop during the growing period. The green and blue VWC of the crops in each county are determined as:

$$VWC_{blue} = \frac{CWR_{blue}}{Y} \tag{4}$$

$$VWC_{green} = \frac{CWR_{green}}{Y} \tag{5}$$

where CWR is crop water requirements (in $m^3/ha$) defined as the crop water evapotranspiration and Y refers to the crop yield (ton / ha).

Using 10-year, county-level data of crop production, the average yield of each crop are calculated. The temporal scale of the applied data set provided by AOSKh [**?** ] of the South Khorasan province is annually and therefore, the assessment of the annual virtual water trade in agricultural production will be possible. Due to the low precipitation in the study region (139 mm annually), the rainfed productions amount is very small (ranging from zero to just 8 percent of total production) and therefore, the major component of total agricultural productions come from irrigated cultivated lands. Regarding the small share of rainfed agriculture, in this study, only the irrigated productions are considered.

## 2.4. Virtual Water Trade in Crops

One of the big challenges in quantifying the virtual water trade in the study region is the lack of data. In most cases, either the imported/exported productions dataset is not available or the access of the researchers to these data sets is limited. In this study, the annual trade in agricultural crops at a county-level scale in the period of 2005 to 2014 has been investigated. Applying the approach proposed in Liu et al. [**?** ], the required amount (tonnage) of each crop for domestic uses is calculated using the per capita consumption of each crop in the region multiplying by the population of each county in the study province. Comparing the required crop amounts in order to provide food security of the inhabitants with the actual (total) crop produced in each county of the region, determine either a particular county will export the surplus productions or must import the crops due to its food shortage.

Due to the lack of the required information of traded water in processed crops in the study region, this amount is not included in the traded virtual water. In addition, it is not possible to trace the final destination of crop consumption in the province level. The main reason for this limitation is the lack of information of transported tonnage of agricultural products alongside the limited access to the available information.

## 2.5. Water Self-Sufficiency and Water Scarcity

The assumption that a water scarce region should rely heavily on importing to reap the benefits from the amount of water imported in order to alleviate the pressure have exerted to their water resources, would be reasonable enough. Regions with copious amount of water resources, though, have this opportunity to take financial advantage of exporting water in the virtual form [**?** ]. To quantify the water scarcity, Water scarcity (WS) index (in some studies is known as the Withdrawal to Availability (WTA) ratio) is defined as the ratio of the water withdrawal (water use) in a region to the total water available in that region [**?** ]:

$$WS = \frac{WU}{WA} \times 100 \tag{6}$$

In which WU refers to the total usage of both rainfall (green) and irrigation (blue) sources of water in crop production and WA is the total available water in a specific region. In this study, the amount of WU is computed by multiplying the calculated virtual water content of each crop (in $m^3/ton$) to the total production of that crop (in ton). The WU in Equation (2) refers only to the amount of water consumed by agricultural practices, and not to other usages of water (e.g., household, industry, mining, services, etc.). Water withdrawal refers to the total water input into the crop production process while water consumption is the fraction of water withdrawal which has removed from the originating basin due to the evapo(transpi)ration, product integration, and discharge into other watersheds [**?** **?** ]. This study focuses on the comparison of the 11 counties of the study province based on the sustainability of the water resources exploitation within their agricultural practices. Due to the severity of the region's aridity, some other sustainability indicators might not contribute to comparable results since most of them are limited to the range of zero to one. Other advanced indicators (such as WSI (Water Stress Index) [**?** ], AWARE (Available Water Remaining) [**?** ], WAVE+ (Water Accounting and Vulnerability Evaluation) [**?** ]) has emerged recently to overcome the limitation of withdrawal to

availability ratio as a characterization factor. For instance, this ratio sometimes shows less scarcity for arid regions compared to water-abundant one's, which do not gain an accurate picture of the situation. However, since this study solely aims to investigate the sustainability of cropping pattern in an arid, and agriculture-based province using the concept of virtual water, applying such indicators do not create a sense of comparison. The reason is that the calculated values of these indicators for most counties in the study region are the same and equal to 1.0, which represents the most unsustainable situation because of their intensive agriculture. In addition, collecting data was the most challenging part of this research, because of the low participation of the local authorities and consequently, the lack of information, such as human water consumption and environmental water requirements, which are required to calculate some of the above-mentioned indicators [**?** ].

With identifying the characteristics of each meteorological station (such as geographical location and the amount of annual rainfall) within each county and using the concept of isohyet lines, WA is computed by using the ArcGis software. The required data to calculate these parameters is provided by Regional Water Company of South Khorasan province. Different situations of a region regarding WS values is shown in Table **??**. As shown in this table, in the case of overexploitation, WS value exceeds 100 percent.

**Table 1.** Categorization of water scarcity [**?** ].

| Water Scarcity (WS) (%) | Situation |
|:---:|:---:|
| WS > 100 | Overexploited |
| $60 \leq$ WS < 100 | Heavily exploited |
| $30 \leq$ WS < 60 | Moderate exploited |
| WS < 30 | Slightly exploited |

To indicate the region's dependency on outside (imported) water resources index of Water Dependency (WD) is defined as the ratio of imported water to the total water consumed in a region [**?** ]:

$$\text{WD} = \begin{cases} \frac{\text{NVWI}}{\text{WU+NVWI}} & \text{if NVWI} \geq 0 \\ 0 & \text{if NVWI} < 0 \end{cases} \tag{7}$$

where NVWI is the net virtual water imported to the region. This index changes in range of 0 to 100 percent in which zero means the region is fully self-sufficient in providing its food requirements and 100 percent refers to a region which imports all its required foods. Therefore, water self-sufficiency (WSS) is inversely related to the water dependency as follows:

$$\text{WSS} = 100 - \text{WD} \tag{8}$$

## 3. Results

In this study, the crops harvested in the province are categorized into six categories as shown in Table **??**. The virtual water contents of crops are calculated in all counties in the province.

**Table 2.** Classifying the different crops used in this research ([*] hereafter only "fiber crops" is used).

| Group (Number of Crops) | Included Crops |
|---|---|
| Cereals (5) | wheat, barley, maize, alfalfa, millet |
| Legumes (4) | chickpea, mung bean, bean, lentil |
| Roots and fiber crops [*] (4) | potato, sugar beet, turnip, cotton |
| Vegetables (9) | tomato, onion, cucumber, eggplant, zucchini, sweet melon, water melon, garlic, cantaloupe |
| Fruits (13) | apple, pear, quince, sour cherry, cherry, plum, peach, apricot, table grape, pistachio, almond, walnut, carrot |
| Oil seeds (2) | sesame, sunflower |

Due to the arid climate and the low annual precipitation (as described in Section **??**), green virtual water (GVW) is a mere fraction of TVW in the study region. The largest proportion of GVW among the six categories belongs to the cereals with only 1.42%. This small contribution of GVW within the province have exerted enormous pressure on the South Khorasan aquifers. Table **??** compares different crop categories regarding their green and blue VWC. As shown in Table **??**, the same crops have different VWC in different counties and therefore, it can be concluded that each county would be suitable for cultivation of particular types of crops. Regarding this issue, during the period of 2005–2014, it could be possible to save more than $346 \times 10^6$ m$^3$ water annually without any reduction in the harvested lands by applying the proper cropping pattern in the region. This huge volume of water could be saved annually if spatial prioritization of crops based on TVW (as shown in Table **??**) had been applied and the cultivation of some type of crops was moved to the counties with the minimum VWC or equally with the maximum water productivity. Table **??** shows that there were remarkable differences in water productivity throughout the province. Using these results in agricultural practice could lead to producing the same amount of crops by using by far less water, and consequently, the considerable pressure exerted on the groundwater could be substantially relieved (column 4 in Table **??**). Unfortunately, this great opportunity for saving a huge amount of water in such an arid region with limited water resources was lost due to ignoring the concept of virtual water by the local authorities.

**Table 3.** 10-year mean of virtual water content and other components in the entire province (ET$_C$: Crop Evapotranspiration, GVW: Green Virtual Water, BVW: Blue Virtual Water, TVW: Total Virtual Water).

| Crops | ET$_C$ (mm) | GVW (m$^3$/ton) | BVW (m$^3$/ton) | TVW (m$^3$/ton) | GVW Proportion (%) | Water Productivity (Kg/m$^3$) |
|---|---|---|---|---|---|---|
| Cereals | 1213 | 49 | 3454 | 3502 | 1.42 | 0.286 |
| Legumes | 981 | 77 | 11,922 | 11,999 | 0.69 | 0.083 |
| Fiber crops | 1464 | 26 | 3293 | 3319 | 0.83 | 0.301 |
| Vegetables | 1026 | 13 | 923 | 937 | 1.36 | 1.068 |
| Fruits | 1563 | 59 | 4994 | 5052 | 1.21 | 0.198 |
| Oil seeds | 1059 | 60 | 13,110 | 13,171 | 0.50 | 0.076 |

The main reasons for high amounts of water footprints in the study region can be as follows: the semi-aridity and climatic situation, the wrong policies of the local authorities for achieving self-sufficiency in providing food, low yields of crops due to the inefficient agriculture, and subsequently, the low values of water used to produce crops.

**Table 4.** Difference in water productivity and the water saving potential.

| Crop Type | Min Water Productivity ($\frac{kg}{m^3}$) | Max Water Productivity ($\frac{kg}{m^3}$) | Difference in Production (%) | Annual Water Saving Potential (Mm$^3$) |
|---|---|---|---|---|
| Cereals | 0.196 | 0.383 | 95.03 | 119 |
| Legumes | 0.049 | 0.152 | 213.32 | 2 |
| Fiber crops | 0.107 | 0.860 | 706.28 | 131 |
| Vegetables | 0.655 | 2.237 | 241.63 | 49 |
| Fruits | 0.161 | 0.269 | 67.34 | 38 |
| Oil seeds | 0.049 | 0.467 | 853.16 | 8 |

The proportion of each type of crops based on their cultivation areas along with the amount of production is depicted in Figure **??**. As shown in this figure, with a share of 7% in cultivated lands, vegetables account for 27% of crop production. At the province-level, cereals are the major type in both cultivated lands (61%) and crop productions (45%). This amount of production has led to export the surplus of 24,117 tonnage annually. It means that $68.31 \times 10^6$ m$^3$ water transferred annually outside the boundaries of the province. However, the South Khorasan province is a net virtual water importer with regard to the other five crop types. Based on the results of this study, these five crop types are responsible for importing $936.87 \times 10^6$ m$^3$ water from outside the province. As shown in Figure **??**, fruits have the largest share of the virtual water imported to the province with $423 \times 10^6$ m$^3$ annually.

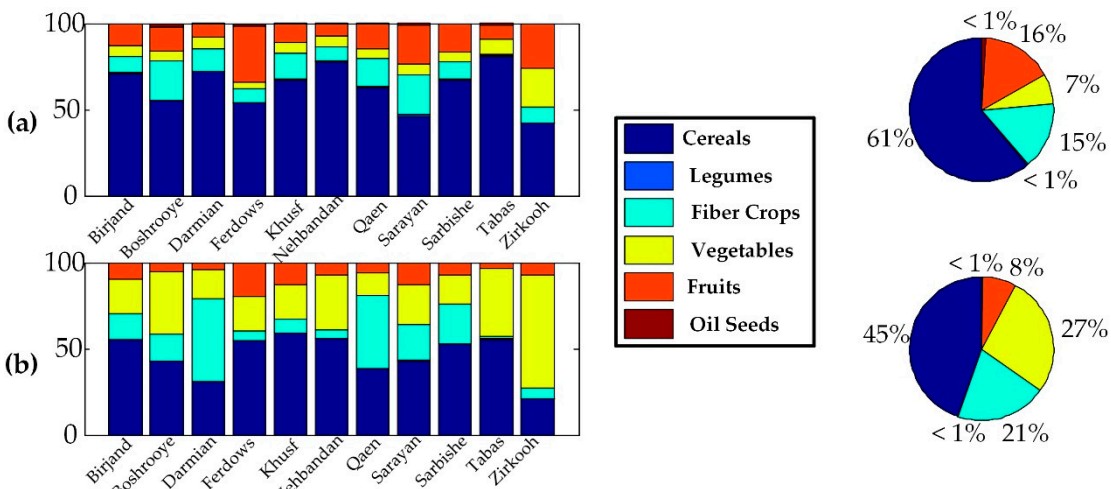

**Figure 4.** (**a**) The state of cultivation area, and (**b**) the crop production in each county.

Analyzing the results of the study shows that the province not only was self-sufficient in producing all types of cereals, but also fulfilled the cereal requirements of some other regions. This situation was neither economically (Figure **??**) nor environmentally sustainable as cereal's virtual water content was relatively high (Table **??**).

During the study period, from 2005 to 2014, there was an inclination towards increasing the cultivation area of the fruits by 52% [**?** ]. The water and agricultural authorities of the province encouraged the farmers to harvest cereals more due to the high economic value of them in comparison with other types of crops (Figure **??**), as well as their high strategic importance. On the other hand, the production of other crop types reduced significantly due to the severe droughts that the whole province confronted with. In spite of these reductions, the pressure on the water resources, mainly includes aquifers, did not decrease. The main reason for this issue is that water and agricultural authorities of the province had no clear idea and understanding from the virtual water concept. Due to this misunderstanding, the production of fruits, as a water-intensive type of crops, increased by

52%. In the period of 2005 to 2014 the remains of crop types, i.e., cereals, legumes, fiber crops, vegetables, and oil seeds, experienced a decline in their harvested area by 24,75,47,28, and 28 percent, respectively. This shift to produce more water-intensive crops increased the groundwater withdrawal from $800.24 \times 10^6$ m$^3$ in 2009 to $1392.62 \times 10^6$ m$^3$ in 2014 (74.02 percent increase).

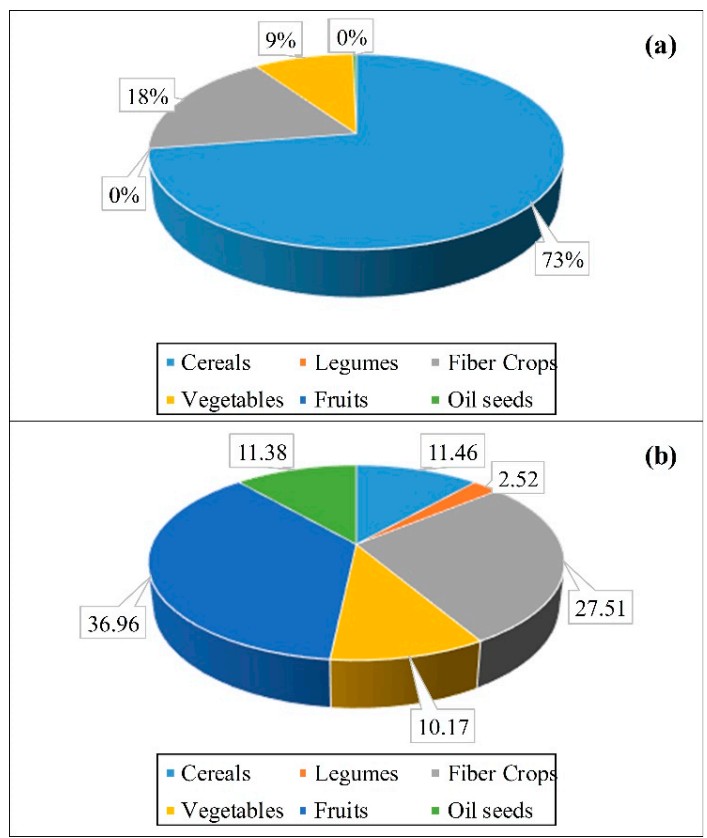

**Figure 5.** (**a**) The fractional contribution of exports, and (**b**) imports of the province during 2005–2014 (1143.53 Mm$^3$ is imported and 274.97 Mm$^3$ is exported at the province level).

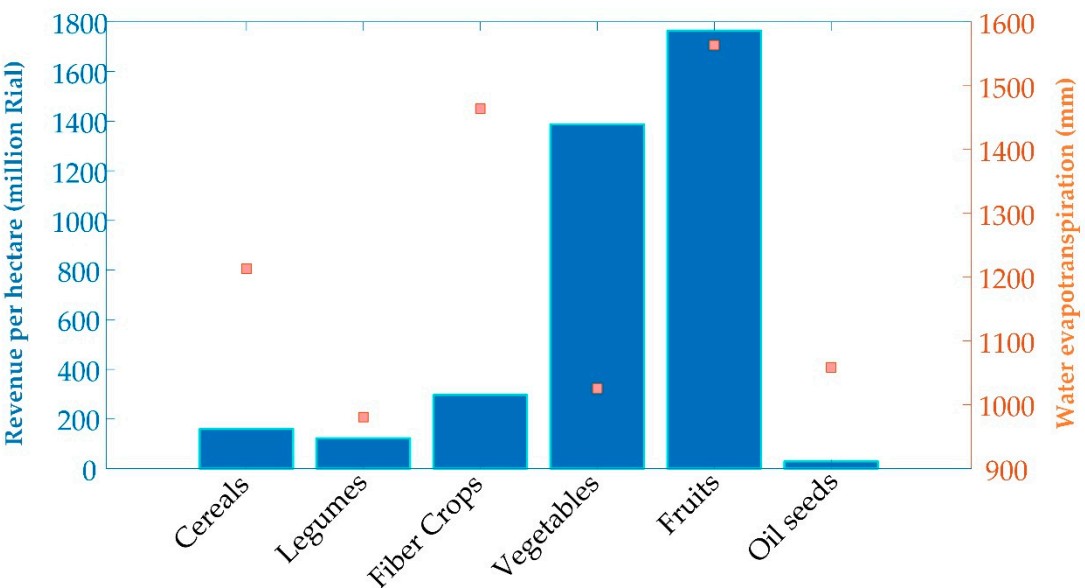

**Figure 6.** 10-year average of South Khorasan crop revenue per hectare and crop water requirement.

Although Boshrooye and Sarayan were almost self-sufficient, this self-sufficiency has been achieved by putting an enormous pressure on their limited water resources and therefore, their WSs were far beyond the bounds of sustainability (Figure **??**). Other counties such as Ferdows, Qaen, Sarbishe, and Zirkuh produced crops unsustainably by a WS of 366, 283, 175, and 171 percent while experienced just 35, 30, 31, and 55 percent of dependency, respectively (Figure **??**). Tabas was slightly exploited due to its small agricultural practices. Khusf, by 49% of WS, utilized its water resources moderately (Table **??**). Birjand, Darmian, and Nehbandan relied heavily on their groundwater resources to achieve the WS of 89, 86, and 71 percent, respectively. Other counties, though, in order to achieve their desired self-sufficiency in agricultural productions overexploited their limited groundwater resources.

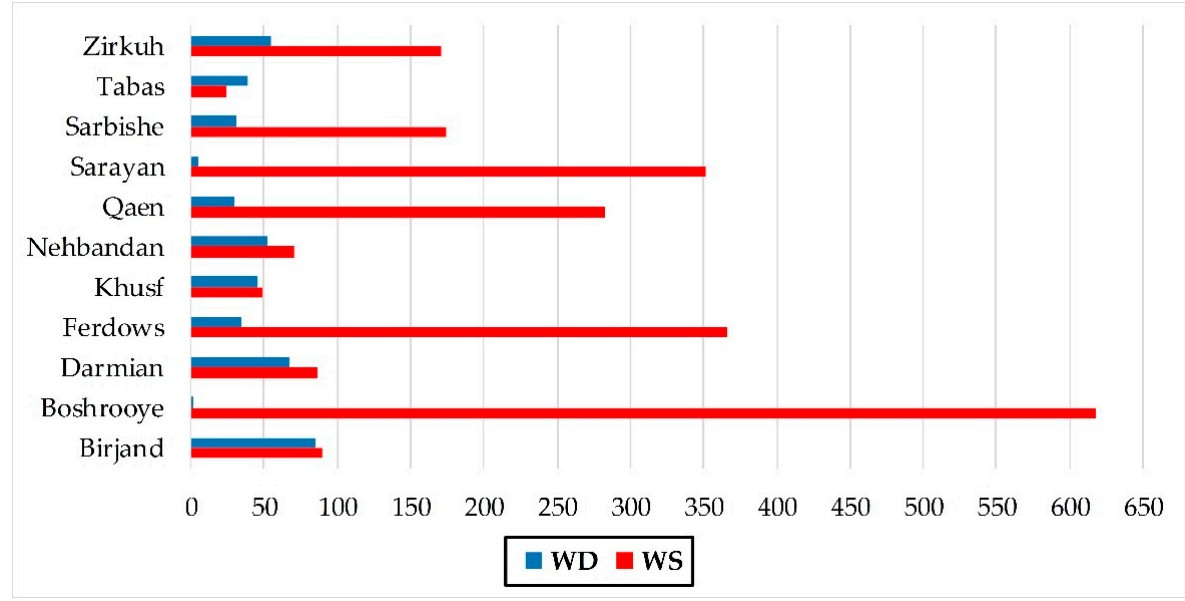

**Figure 7.** Water dependency against water scarcity in each county.

The contribution of available renewable water resources of each county is quantified using the isohyet lines from the long-term precipitation data. Unfortunately, the data of water availability for the period of 2005 to 2010 was not available in the Regional Water Company of South Khorasan province, and therefore, only the available data for the period of 2011 to 2014 was used in this study. The amount of water withdrawal along with the available water of 11 counties of the province is presented in Table **??**. According to this table, Boshrooye had the largest water usage with the average of $221.43 \times 10^6$ m$^3$ annually. Qaen and Sarayan were the next largest water users with an average withdrawal of $195.51 \times 10^6$ m$^3$ and $173.17 \times 10^6$ m$^3$, respectively. On the other side, Khusf and Tabas used the least amount of water in crop productions with $28.72 \times 10^6$ m$^3$ and $31.05 \times 10^6$ m$^3$, respectively.

A similar trend of WSs in each county within the province was identified. As illustrated in Figure **??**, an increasing trend in WS found for all the counties, excluding Birjand where its WS fluctuated from year to year. Ferdows, though, experienced an abrupt change in its WS in 2014 as its cultivation area dramatically increased from 4329 hectare in 2013 to 19,637 hectare in 2014. This increasing trend is mainly due to misunderstanding of the food security concept by the policymakers. They translated the "food security" as producing all the food requirements within Iran's territory to reduce dependency to foreign countries or even achieving absolute self-sufficiency, regardless of long-term consequences of unsustainability. With this incorrect definition of food security, the consequent water resources problems in the region would be expectable.

**Table 5.** Water withdrawals and availabilities for each county during 2011–2014.

| Counties | Water Withdrawal ($10^6$ m$^3$) | | | | Water Availability ($10^6$ m$^3$) | | | |
|---|---|---|---|---|---|---|---|---|
| Year | 2011 | 2012 | 2013 | 2014 | 2011 | 2012 | 2013 | 2014 |
| Birjand | 28.72 | 26.20 | 25.71 | 22.10 | 31.12 | 31.11 | 24.72 | 25.44 |
| Boshrooye | 160.29 | 208.46 | 216.39 | 232.17 | 39.71 | 39.69 | 31.55 | 32.47 |
| Darmian | 32.94 | 31.67 | 31.25 | 44.15 | 47.39 | 47.37 | 37.65 | 38.75 |
| Ferdows | 67.72 | 86.14 | 81.29 | 263.27 | 39.12 | 39.10 | 31.08 | 31.99 |
| Khusf | 56.43 | 64.92 | 51.72 | 63.66 | 134.12 | 134.07 | 106.56 | 109.66 |
| Nehbandan | 84.70 | 101.56 | 126.10 | 109.73 | 168.62 | 168.56 | 133.97 | 137.87 |
| Qaen | 181.09 | 180.53 | 176.13 | 183.98 | 69.92 | 69.89 | 55.55 | 57.17 |
| Sarayan | 102.78 | 158.40 | 160.94 | 159.52 | 46.39 | 46.37 | 36.86 | 37.93 |
| Sarbishe | 80.11 | 106.16 | 104.64 | 115.73 | 64.70 | 64.68 | 51.41 | 52.90 |
| Tabas | 57.84 | 83.03 | 83.47 | 86.12 | 355.02 | 354.89 | 282.06 | 290.29 |
| Zirkuh | 91.18 | 99.29 | 102.47 | 98.42 | 63.65 | 63.62 | 50.57 | 52.04 |

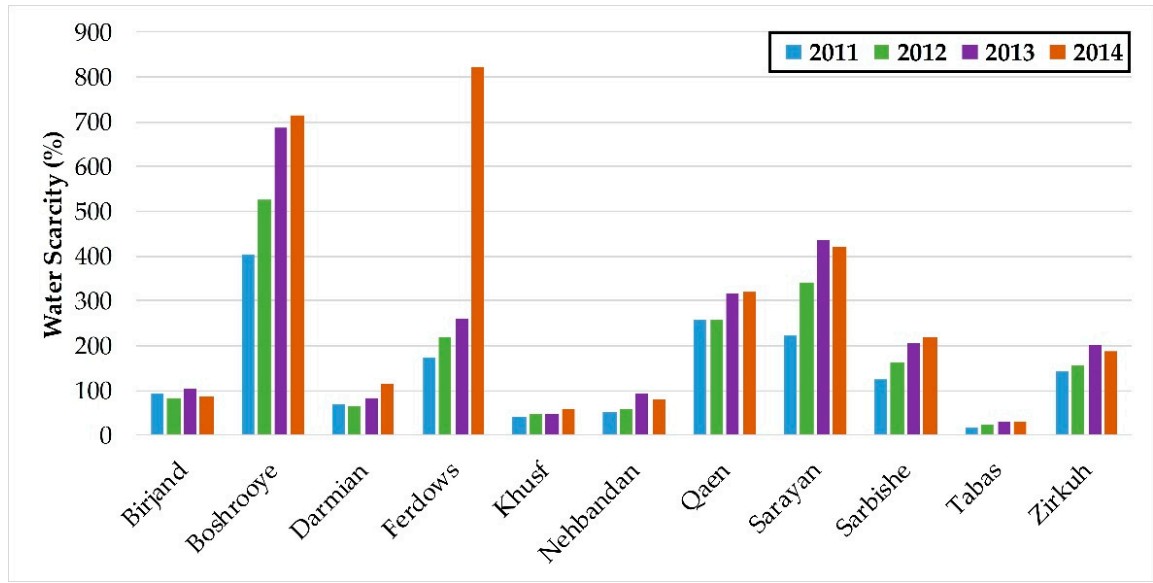

**Figure 8.** Temporal variability of the water scarcity (WS) in all counties from 2011 to 2014.

The proportion of crops transferred within the province are depicted in Figure **??** either the crops imported to or exported from the counties. Cereals were the largest type of crops which either exported from or imported to the counties. Birjand, Darmian, and Zirkuh imported around 65.7% of the cereal productions of other counties annually and the remains were exported outside the province, including about 24,117 tonnage of cereals which is equivalent to $68.31 \times 10^6$ m$^3$ of water. According to the values of WS of the counties, as shown in Figure **??**, this situation was not environmentally sustainable. In other types of crops the province was net virtual water importer and all the virtual water exports in them were interregional. This amount of imported water was not enough to alleviate the enormous pressure exerted on the groundwater aquifers as the average WS in the South Khorasan province was 206%.

Figure **??** provides a comparison between water evapotranspiration and average revenue per hectare of different crop types. From 2005 to 2014, the total water that used in the production of cereals, legumes, fiber crops, vegetables, and oil seeds decreased by 25% or $217.06 \times 10^6$ m$^3$. The irrigation water consumed in fruits, though, increased dramatically by 148% or $251.00 \times 10^6$ m$^3$. This reallocation of limited water resources to water-intensive crops like fruits resulted in increasing water withdrawal

during this period (Figure **??**). Regarding the depicted results in Figure **??** along with the results shown in Table **??**, vegetables are the most environmentally (low TVW) and second economically (high revenue per hectare) sustainable category. Therefore, it is worth to increase the cultivation area of this crop type. By applying this crop pattern, farmers would gain more profit by using the same amount of water or even less.

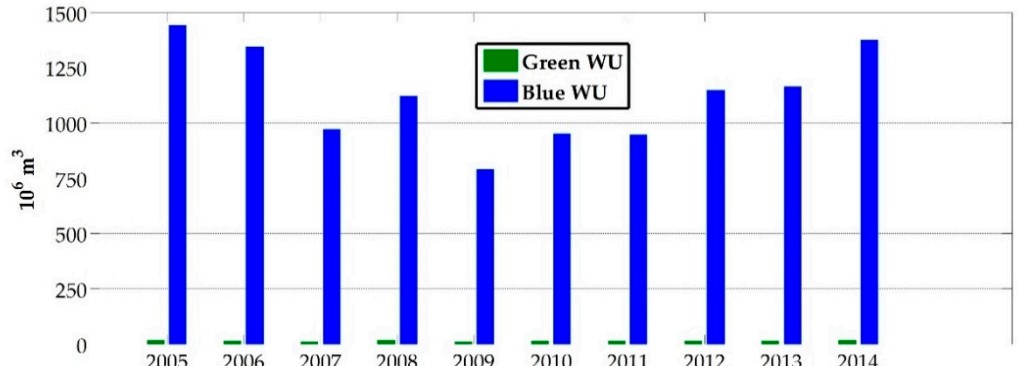

**Figure 9.** The variation of green and blue water use (WU) in the province from 2005 to 2014.

As can be seen in Figure **??**, fruits have the biggest evapotranspiration (1563 mm per year), followed by fiber crops (1464 mm per year). Legumes and vegetables have the minimum values of evapotranspiration with 981 and 1026 mm per year, respectively. Although the crop water requirement of legumes and oil seeds are much lower than the other categories, but their low yields lead to the high virtual water contents with 11,999 $m^3$/ton and 13,171 $m^3$/ton, respectively. Vegetables with 937 $m^3$/ton are ranked the latest place in respect with TVW.

Fruits, however, is the type that captured the greater amount of imports with $422.61 \times 10^6 \, m^3$ annually. Fiber crops ($265.33 \times 10^6 \, m^3$), oil seeds ($129.20 \times 10^6 \, m^3$), vegetables ($90.88 \times 10^6 \, m^3$), and legumes ($28.84 \times 10^6 \, m^3$) held the next positions, respectively.

Birjand and Darmian, although, were net virtual water importers in all crop types, but their average WS of 92% and 83%, respectively implies that their crop production were not sustainable. This was also true for other counties as well, excluding Tabas with 25% and Khusf with 49% of WS. The low WS in Tabas was mainly due to its large territory (accounted for 37% of entire province's area) in comparison with other counties and its low harvested area as a result of its harsh climatic condition. Other counties, though, imported some type of crops while exported the other ones. The contribution of each type of crops is presented in Figure **??**.

In terms of sustainability, nine counties put their valuable and limited groundwater resources in grave danger of deprivation. Ferdows, Boshrooye, and Sarayan continued to rely heavily on their groundwater aquifer systems as a result of self-sufficiency policies without considering the limited water resources (Figure **??**). South Khorasan province has suffered the consequences of this mismanagement. Drying up lakes and wetlands, land subsidence and sinkholes, water quality degradation, desertification, coming down the groundwater levels, soil erosion, and dust storms as environmental consequences and losing lots of jobs, migration to towns, and suburban sprawl as social ones were prompted as a result of the condition of insecurity [**?** ].

## 4. Conclusions

In the study region (South Khorasan province, Iran) due to the low precipitation and high rates of evapotranspiration, green virtual water (GVW) accounts for a small fraction of total virtual water (TVW). Besides, in this occasion, the national policy to achieving full self-sufficiency in producing all the agricultural produce has exerted enormous pressure on groundwater aquifers. In line with this policy, overexploitation has occurred in most counties of the South Khorasan province. The results of this study show that there are significant differences in the productivity of same crop categories in

different counties throughout the province mainly due to different climatic parameters. It is shown that if the spatial prioritization for harvesting different crop categories had been established by the local authorities, more than $346 \times 10^6$ m$^3$ water could be saved annually in the agricultural practices within the province without any reduction in cultivation area. Although there was a considerable decrease in the cultivation area of some crops during the period of 2005–2014, this reduction in cultivation areas did not contribute to water saving. This was largely caused by the expansion of water-intensive crops cultivation, namely fruits.

Some useful indicators such as Water Self-sufficiency (WSS), Water Scarcity (WS), Water Productivity, and the revenue per one unit of cultivation area used in this study in order to assess the sustainability of the agricultural practices applied in the South Khorasan province during the period of 2005–2014. In addition, the annual changes in Water Scarcity of counties within the province have been quantified in this study. The results show that almost all the counties are heavily exploited or even overexploited. Generally, there was an increasing trend in Water Scarcity of counties due to the growth of cultivation of water-intensive crops, such as fruits, which led to an increase in water exploitation.

Considering the water-related problems in South Khorasan province, there is a crucial need to change management strategies for limited available water resources. For example, importing water-intensive crops instead of producing them in the province can alleviate the situation. Furthermore, the region authorities should understand the food self-sufficiency concept correctly and, consequently, the meaning of the "development". Unfortunately, the current agricultural practices in the province is neither environmentally nor socioeconomically sustainable. Importing foods from outside the borders of the province on a national scale instead of producing them locally can save a huge amount of water annually. The results of the study indicate that, from a sustainability point of view, importing cereals and fruits are of high priority not only due to their high consumption in the province but also for their high TVW. Improving the water productivity, establishing the proper cropping pattern, and facilitating the imports of water-intensive crops would be beneficial in achieving food security rather than sticking firmly to current food policy in which domestic food production is of the most importance. Future researches are needed to take into account all the agricultural produce and the total water engaged in both production and consumption chain. The methodology applied in this study can be used in any other regions to evaluate the sustainability of their water resource consumption.

**Author Contributions:** All authors contributed extensively to the work presented in this paper. As specific contributions, E.Q. contributes in Conceptualization, Data curation, Formal analysis, Methodology, Validation, Visualization, and Writing—original draft. A.A. contributes in Conceptualization, Methodology, Project administration, Supervision, Validation, and Writing—review & editing.

**Acknowledgments:** The authors acknowledge the South Khorasan Province Metrological Organization (SKhMet) and the Regional Water Company of South Khorasan (RWC-SKh) for their collaborations on providing the data and the information used in this study.

**Conflicts of Interest:** The authors declare no conflict of interest.

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
