# Peer review of "Assessment of Agricultural Water Resources Sustainability in Arid Regions Using Virtual Water Concept: Case of South Khorasan Province, Iran"

_water, doi:10.3390/w11030449_

Round 1

Reviewer 1 Report

This is a study looking at water resource sustainability using water footprint (WF) / virtual water content concepts (VWC).  Although there are differences of WF and VWC   the authors used the two concepts without making a clear and reasonable connection between them.

Introduction can be improved by expressing some references of WF /VWC from Iran crops /cultivation systems, Water self-sufficiency cases etc.  

Results have to reorganized. The reader has to follow text near with tables / figures in comment.

A conclusion section /paragraph is needed to summarize basic findings and make an overall evaluation of the method used for a better water resource sustainability management.

Authors can find specific comments in the attachment manuscript

Author Response

All the reviewer's comments are responded point-by-point in the attached file.

Reviewer 2 Report

A massive work has been done and therefore I recommend it for publication in its present format

Author Response

(The authors gave the same response as above.)

Round 2

Reviewer 1 Report

The manuscript has been improved significantly and I recommend  its publication after 3 minor revisions that can be found in the text.

Author Response

The reviewer's comments have been addressed point-by-point in the attached file (2nd Round)
